# Kids in a Candy Store: An Objective Analysis of Children’s Interactions with Food in Convenience Stores

**DOI:** 10.3390/nu12072143

**Published:** 2020-07-18

**Authors:** Christina McKerchar, Moira Smith, Ryan Gage, Jonathan Williman, Gillian Abel, Cameron Lacey, Cliona Ni Mhurchu, Louise Signal

**Affiliations:** 1Department of Population Health, University of Otago, Christchurch 8140, New Zealand; jonathan.williman@otago.ac.nz (J.W.); gillian.abel@otago.ac.nz (G.A.); 2Health Promotion and Policy Research Unit, Department of Public Health, University of Otago, Wellington 6242, New Zealand; moira.smith@otago.ac.nz (M.S.); ryan.gage@otago.ac.nz (R.G.); louise.signal@otago.ac.nz (L.S.); 3Māori and Indigenous Health Institute, Department of the Psychological Medicine, University of Otago, Christchurch 8140, New Zealand; cameron.lacey@otago.ac.nz; 4National Institute for Health Innovation, University of Auckland, Auckland 1142, New Zealand; c.nimhurchu@auckland.ac.nz

**Keywords:** food availability, food marketing, childhood obesity, convenience stores, wearable cameras

## Abstract

Increasing rates of childhood obesity worldwide has focused attention on the obesogenic food environment. This paper reports an analysis of children’s interactions with food in convenience stores. Kids’Cam was a cross-sectional study conducted from July 2014 to June 2015 in New Zealand in which 168 randomly selected children aged 11–14 years old wore a wearable camera for a 4–day period. In this ancillary study, images from children who visited a convenience store were manually coded for food and drink availability. Twenty-two percent of children (*n* = 37) visited convenience stores on 62 occasions during the 4-day data collection period. Noncore items dominated the food and drinks available to children at a rate of 8.3 to 1 (means were 300 noncore and 36 core, respectively). The food and drinks marketed in-store were overwhelmingly noncore and promoted using accessible placement, price offers, product packaging, and signage. Most of the 70 items purchased by children were noncore foods or drinks (94.6%), and all of the purchased food or drink subsequently consumed was noncore. This research highlights convenience stores as a key source of unhealthy food and drink for children, and policies are needed to reduce the role of convenience stores in the obesogenic food environment.

## 1. Introduction

Increasing rates of childhood obesity worldwide are a major public health concern [1]. In 2016, it was estimated that 50 million girls and 74 million boys were obese and that a further 213 million children were overweight, increasing their risk of noncommunicable disease over the life course, especially type 2 diabetes [2]. Research attention has therefore focused on the obesogenic food environment and how it influences dietary behaviour and bodyweight in children [3,4,5,6,7]. New Zealand children have the second highest rates of obesity in countries affiliated with the Organisation of Economic Cooperation and Development (OECD) [8], and this is patterned by socioeconomic deprivation and ethnicity [9]. New Zealand children grow up in obesogenic environments, of which convenience stores are a feature [10]. Food retailers, including convenience stores, feature prominently in New Zealand (NZ) children’s lives [11].

Convenience stores are a unique setting; the food and drinks available are overwhelmingly unhealthy and ultra-processed [12,13]. They are often located within children’s neighbourhoods, especially around schools. In many Western-developed countries, there is a higher density of convenience stores in close proximity to schools in socioeconomically deprived areas [10,14,15], and in some countries, especially in South Korea and Japan, the number of convenience stores have increased as part of the overall all food environment [16,17]. This is of concern, as the proximity and density of convenience stores in a child’s neighbourhood, including their home and school neighbourhoods, is positively associated with unhealthy eating behaviours [18,19] and overweight [7,20].

The association between the proximity of food outlets to food purchases and behaviour are the subject of some debate, as research does not always take into account food outlets on commuting routes [21] or that people travel for food [22]. This impact may be limited for children, as they are less likely to travel independently and their movements tend to be localised to a distinct geographic area [11]. Investigating the consumer food environment is inherently complex [23]. Research has typically measured the healthfulness of a food store through checklists or measurement of the shelf-space for key indicator foods [23,24]. Food purchases have been measured through the collection of receipts [25] and food consumption by food recall methods [19]. A limitation of these methods is bias from both participant recall and social desirability [26]. It is rare that research has measured both the characteristics of a store as well as a child’s purchasing and behaviour within the same study time period; therefore, the need for research to better measure this interaction has been identified [4,7].

Wearable cameras are a relatively new research tool. They have been used to document children’s health behaviours related to the environment in which they live [27,28]. The innovative Kids’Cam New Zealand (NZ) study [29] has enabled objective analysis of the world in which children live, including the availability of drinks in children’s lives [30] and children’s exposure to food and alcohol marketing [31,32]. The FoodSee methodology was developed to analyse people’s interaction with the in-store food environment [33]. Using the FoodSee methodology with the Kids’Cam data, this study aimed to examine children’s interactions with convenience stores. In this paper, the term interaction is used to define aspects of the consumer food environment within a store specifically: food and beverage availability, and marketing together with purchasing and consumption behaviour. The objective measurement enabled the research to overcome the limitations of previous food recall methods.

## 2. Materials and Methods

Kids’Cam NZ [29] was a cross-sectional study conducted from July 2014 to June 2015 in the Wellington region of NZ in which 168 randomly selected children aged 11–14 years old (year 8) wore a wearable camera for a 4-day period (Thursday to Sunday). The camera captured 136° images of the children’s surroundings every 7 seconds, generating approximately 1.3 million images. The sampling strategy resulted in representation from children with a range of ethnic and socioeconomic backgrounds from schools that were widely distributed throughout the three main areas of the Wellington area—Wellington City, Porirua and the Hutt Valley. The total number of children in year 8 in Wellington for 2014 was 4883. Ethical approval was given by the University of Otago Human Ethics Committee (Health) (13/220) in 2013 to analyse the data for any issue of public health interest. Therefore, the children were blinded to the specific aims of the Kids’Cam NZ studies, including the current study. Further details of the methods used for Kids’Cam NZ have been published previously [29,31].

In this ancillary study, “Kids’Cam Convenience Stores”, 37 children who had image data showing the inside of a convenience store or service station were eligible for inclusion. Service stations were included as they are a source of energy-dense snack foods with limited healthy options [34,35] and it is likely that children may use service stations in a similar way to a convenience store. Convenience stores and service stations were defined by signage as well as a limited number of ≤2 checkouts [33].

### Coding and Data Analysis

Each of the images from a convenience store visit was manually coded according to the FoodSee study protocol, which was piloted and refined by C.M. and M.S. [33]. The study protocol and definitions are available at https://www.otago.ac.nz/heppru/research/index.html. The aspects of the convenience store environment coded for were food availability, accessibility and marketing from the images from each store visit by a child. The aspects of child’s behaviour measured were food purchase and consumption.

Availability was measured by counting the number of individual unique food and drink items in each image. For example, each individual chocolate bar product in an image was counted. Care was taken to ensure each item was only coded once per visit. Two coders (C.M. and M.S.) carried out a reliability test by coding 10 subsequent images from a visit to ensure there was concurrence in the count [33]. The items were also coded for product category based on those used in previous Kids’Cam studies and were modified to reflect the convenience store environment, e.g., by accounting for the different varieties of confectionary sold including single-serve confectionary (confectionary such as lollipops sold as an individual item) and lolly-mixtures (individual candies grouped into bags), and chocolate, which included chocolate products such as single serve bars or chocolate products such as Easter eggs [33]. Product categories also included “iced confectionary” (ice cream or ice blocks/lollies); “snack foods” (potato crisps); “cookies, cakes and pastries” (cakes, muffins, sweet biscuits, sweet and savoury pies, sweet and savoury pastries, slices, scones and sausages rolls); and “sugar-sweetened beverages (SSBs) and fruit juices”. These included carbonated beverages and soft drinks including sports drinks, energy drinks, flavoured milks (chocolate milk), fruit drinks, powered drinks (Milo and Raro), cordial, fruit juices including 100% fruit juices, iced tea, breakfast drinks and flavoured waters. Diet drinks including flavoured waters with noncaloric sweeteners were not included in the sugary drinks category but were coded separately as “Diet drinks”. For analysis, foods and drinks were then categorised into healthy (core) or unhealthy (noncore) using a WHO nutrient profile specific to children [36]. Core foods were from the following categories: milk and milk products with <10 g / 100 g sugar; water; breads and cereals with <15 g / 100 g sugar; fruits and vegetables, including dried fruit; and meat and alternatives such as meat/eggs/nuts, including nut products such as peanut butter but excluding processed meat products, e.g., ham or beef jerky. Other food items were categorised as noncore.

The placement of food and drink items within the convenience store was noted, with the position of the food and drink coded as “accessible” if it was within easy reach of the child. As the images were taken from a camera at a child’s chest height, products at the forefront of an image were considered “accessible”; however, those that were placed in high shelves were considered inaccessible (mean height = 1.59 m, SD 0.075 m) [33]. The position of food and drink items relative to the countertop was also coded. Aspects of food and drink marketing coded for included product packaging, price promotions, branded displays and promotional signs. If an item was packaged and its “brand” name was visible, it was coded. Visible price promotions for products were also coded; for example, Figure 1 has an image with two items for NZ 0.30 c. The number of branded displays which included fixtures supplied by a manufacturer to display their products were coded, for example, promotion boxes containing chocolate as well as branded ice-cream freezers or branded drinks refrigerators, as shown in Figure 1. Promotional signs such as posters were also coded. “Food or Drink Purchases” were coded if a financial transaction occurred at a shop counter in exchange for a food or drink item. Food or drink consumption was coded if a sequence of images then showed the food or drink item being consumed.

A Microsoft Excel spreadsheet was used for the coding of image data. Excel and Stata 14 (Statacorp, College Station, TX, US) were used for descriptive statistical analysis. Children’s demographic characteristics, types of food and drink available, marketing exposure, and purchase and consumption behaviours were summarised using descriptive statistics (counts and percentages for categorical data, and means and standard deviations or medians and interquartile ranges for continuous data). Means and 95% confidence intervals (CI) were calculated for core/noncore food and drink availability. The difference in core/noncore marketing exposure was calculated for each child and summarised as means with 95% CI.

Therefore, this analysis details the aspects of the convenience store environment experienced by a child through the indicators, food availability, accessibility and marketing. The analysis also describes what was purchased and consumed. These indicators combined together are used to describe the children’s interaction with the convenience store environment in this study.

## 3. Results

Thirty-seven children or 22% of the Kids’Cam NZ participants collected image data that showed a convenience store visit. There were 168 visits in total (see Table 1). Most children who visited a convenience store did so only once during their four-day data collection period. A small number visited more than once, with the maximum number of visits being six by one participant). In all, 38 individual stores were visited, with six more than once.

To identify convenience stores, Google Street View was used to determine if the stores were different or if the children were visiting a small number of the same stores repeatedly. The images of each visit were analysed for distinguishing geographical features, such as signage with the name of the store or nearby street names. Thirty-eight different convenience stores or service stations were identified. There were not enough data to identify five of the convenience stores or service stations. Thirty-two stores were visited once; however, six stores were visited more frequently, with the most visited store frequented six times by two different children. Nevertheless, there was enough variation in the sample to combine the data from each convenience store visit to provide information to determine the mean of the different food and drink types available to each child. Analysing child-level data rather than store-level data was in keeping with the study aim to examine children’s interactions with convenience stores.

The images showing convenience store visits were a small proportion of the total number of images collected by each child. All children theoretically had the opportunity to visit a convenience store in the hours that they did not attend school; however, not all children had complete data over the four-day period. The proportion of usable images collected from convenience stores or service stations divided by the total number of usable images for each participant not including school hours was 0.37%. This small proportion indicates that, of approximately 1.4 million images collected overall, there were 11 images, on average, for each convenience store visit [33]. Of the children who did collect usable image data for the current study, their demographic characteristics were comparable to the overall Kids’Cam NZ sample in terms of gender, age, ethnicity, BMI or school decile stratum, as shown by Table 2.

### 3.2. Types of Food and Drink Items Available

Noncore food and drink items dominated the food and drink available to children in convenience stores at a rate of 8.3 to 1, as illustrated in Figure 1 and Figure 2. A mean of 300 (95%CI 220–389) noncore food and drink items and 36 (95%CI 19.2–53.2) core items were available to each child across convenience store visits. The most commonly recorded noncore category available was confectionary with a mean of 160.2 items per child, followed by sugary drinks with a mean of 88.4 and snack food including potato crisps with a mean of 44.6. The confectionary category was dominated by single-serve confectionary with a mean of 65.1 items per child followed by chocolate (58.1) and lolly-mixes (15.8). Other noncore foods available included cookies, cakes and pastries (14.0) and iced confectionary (3.3). The category “Other noncore food or drinks” (e.g., butter and cream) were available at a mean of 8.2 items per child. Of the 36 core food and drink items available to children, on average, the most common categories were fruits and vegetables (included canned or frozen products); meat and alternatives; and bread, milk and water.

### 3.3. Placement of Food and Drink Items by Category

The placement of food and drink items available within convenience stores differed by food and drink category. As shown in the top left image of Figure 3, there was a substantial number of confectionary items, especially chocolate and single serve confectionary and lolly-mixes, on or beside the countertop and directly underneath the counter at heights accessible to children. A mean of 16.9 items of single-serve confectionary, 9.9 items of chocolate and 8.0 items of chewing gum were available on the counter per visit. The placement of these food items required children purchasing a food item to reach over single-serve confectionary food items placed on the countertop.

### 3.4. Promotion

The 37 children who visited a convenience store were exposed to marketing for noncore food and drinks through a variety of mediums. As shown in Table 3, there was a mean of 7.9 exposures per child to noncore packaging, 6.8 exposures to noncore signs, 4.2 exposures to noncore branded displays and 2.8 exposures to noncore price promotions. This was substantially higher than exposures across all mediums to marketing for core food and drinks, with mean differences between core and noncore exposures of 6.1 (95%CI 4.9–7.4) for packaging, 5.4 (95%CI 4.0–6.8) for signs, 3.6 (95%CI 2.8–4.5) for branded displays and 2.7 (95%CI 2.0–3.4) for price promotions.

Most food and drink items in the convenience store environment are packaged, and this package often includes a branded promotion, for example, a logo. Commonly packaged products were confectionary, followed by sugary drinks. Branded displays were also used to display items, especially individual chocolate bars, cookies that were displayed in boxes for individual purchase or lollipops displayed on counters. Chewing gum was displayed in a branded display usually on countertops. Sugary drinks were displayed in branded fridges, and iced confectionary was displayed in branded freezers. The noncore food and drink categories also dominated the items promoted by pricing, in particular, confectionary, which was typically available for very low prices, e.g., NZ 0.10c. In each convenience store, there were also several signs promoting food and drinks, especially chocolate, iced confectionary products and cookies.

### 3.5. Food or Drink Purchase and Consumption

Overall, 70 food or drink items were purchased during children’s convenience store visits (see Figure 3). The purchases were made by the participant or by a companion of the participant who were children of a similar age or an adult. The majority of items purchased were noncore foods or drinks (94.6%); only four core food items were purchased (see Table 4). The most frequently purchased food and drink category was confectionary which, when combined with lolly-mixes and chocolate, numbered 33 items, followed by sugary drinks (20) and ice-cream (11). There were 33 separate instances of food or drink consumption by participants following purchases at a convenience store, all of which were noncore, most frequently confectionary and sugary drinks.

## 4. Discussion

In this study, over one-fifth (22%) of the sample (*n* = 37) visited convenience stores on 62 occasions during their 4-day data collection period. They encountered, on average, 300 noncore food and drink item/child, at a rate 8.3 times that of core food and drink items (mean 36 core food and drink items/child). Most of the noncore food and drink available was confectionary items, especially single-serve confectionary, chocolate and sugary drinks. Snack foods, including potato crisps and packaged single-serve cookies, were also widely available. These findings are consistent with previous research that has found that the product assortment in convenience stores is dominated by noncore food and drinks [12,38].

Sanders-Jackson et al. found that half the US adolescents (13–16y) in their sample visited a convenience store at least once a week [39]. Our study did not include data collection over the week, and the participants in our study were younger. However, it is concerning that more than one in five (22%) children in our sample visited a convenience store during a 4–day period, some more than once. Moreover, we also know that the children in the Kids’Cam NZ study visited other stores where noncore food is sold, such as fast food outlets, and do so regularly [11].

The noncore foods and drinks the children in this study encountered were promoted through a variety of mechanisms, including placement, price promotions, packaging, branded displays and signage [40,41,42]. The placement of single-serve confectionary directly on the countertop or immediately underneath or beside the counter was especially conspicuous. When children were paying for a food item, they often had to reach over single-serve confectionary food items placed on the countertop. In addition, the confectionary displayed underneath the countertop featured prominently in the images were proximal to a child’s chest height and therefore could be easily reached.

In a previous analysis of the Kids’Cam data that measured marketing to children, it was found children were exposed to 27 noncore and 12 core marketing exposures per day [31]. Exposures in convenience stores and supermarkets were excluded from the analysis as the marketing examples were too numerous to count [31]. In the current study, children who visited a convenience store had a mean of 7.9 exposures to packaged noncore products, 4.8 exposures to noncore branded displays and 6.8 exposures to noncore signs. There was a significant difference across all marketing mediums between exposure to noncore and core food and drink marketing in convenience stores. Therefore, the children in the Kids’Cam study who visited convenience stores had greater overall marketing exposure to noncore food and drinks.

Noncore food and drink items were also the dominant food and drinks promoted by pricing at a mean rate of 2.8 items per child. For example, single-serve confectionary was often promoted at low prices (at NZ$1 or less), as seen in Figure 3. International evidence shows that the low price of food at convenience stores is appealing to children [43,44].

Almost all of the food purchased (*n* = 70) and all the food consumed by the participants (*n* = 33), was noncore, which in turn substantially contributes to children’s overall energy intake [43] This finding is consistent with other studies [19,25,45] that have associated convenience stores with unhealthy food purchase and consumption. Although we did not determine the motivation behind children’s food purchasing decisions, it is possible that the prominent marketing and availability of noncore food and drink in convenience stores was a contributing factor.

To our knowledge, this is the first study to objectively measure the food and drink available and marketed to children in convenience stores in their everyday life and with the children’s food and drink purchase and consumption. The use of wearable cameras overcame previous methodological limitations, such as the use of food diaries or dietary recall to measure consumption or the collection of receipts to measure food purchase. The methodology enabled the food and drink available and marketed to children in a convenience store to be measured from their perspective and to observe their subsequent behaviour.

While this research provides evidence on the food and drink available to children in convenience stores and their food and drink purchase and consumption, it has some limitations. The study was cross-sectional in design; therefore, only associations can be drawn from the data. The relatively small sample size of children and the geographical coverage of the study limits the generalizability of the results. The method could not determine if a child saw the food and drink that was available in the image. It is possible that they were looking in a different direction. However, given the amount of noncore food and drinks available in convenience stores, it is likely the children could see noncore food and drinks not captured in the images. It is also likely that the amount of food and drink available was underestimated, as some items were unable to be coded. For example, iced confectionary in freezers was not visible unless a child was directly standing over a freezer as was single-serve confectionary stacked underneath other items in a box on a counter. Future research could use the FoodSee methodology to describe children’s interactions with other food settings such as supermarkets or takeaway outlets. Future research could also increase the sample size so that specific ethnic or socioeconomic differences could be described.

This research focuses on the convenience store setting as a source of unhealthy food and drink in children’s lives that contributes to unhealthy food and drink consumption. To reduce and prevent childhood obesity, strategies to limit the impact of convenience stores need to be considered. Policy options include limiting the density and proximity of convenience stores located near schools such as in South Korea where green food zones, which ban convenience stores within 200 m of a school, have been implemented [45,46].

There is international evidence on programmes that involve working with convenience store owners to improve the range of healthy food and drinks that are available and promoted [38,47] and to reduce the availability and promotion of unhealthy food and drink products, such as removing them from countertops and areas above and below the cash registers (at checkout) and end-of-aisle shelves [48]. Sugar-sweetened beverages could be grouped together in a common aisle and section of the coolers to allow people to more easily identify them and to avoid them. Other such strategies could include limiting the price promotion of unhealthy food and drinks and reducing branded displays [47]. However, these strategies would be challenging in the NZ environment as convenience stores are mostly independently owned. National policies, such as introducing plain packaging [49] or the introduction of a tax on sugary drinks, would likely impact purchasing behaviour and dietary intakes [50]. Confectionary was the major noncore food and drink category available and purchased. Policy makers should also consider strategies to limit its availability and promotion alongside those that limit sugary drinks.

Convenience stores are located centrally within communities and, in many societies, are an important food source for people with limited mobility due to age, income or disability [16,47,51]. Given that substantial changes to the in-store environment would need to occur in order to impact food and drink purchasing behaviour of children [25], future research could also explore the role of convenience stores in the community. Communities have the right to a local source of healthy food options, and it is time to question if the current food retail environment is delivering this.

## 5. Conclusions

This research highlights convenience stores as a key source of unhealthy food and drink for children and associates their purchase and consumption of unhealthy food and drink products to this setting. It also measures aspects of the in-store environment from the child’s perspective and describes the myriad of promotion strategies used within convenience stores to encourage unhealthy food and drink purchases. The findings provide evidence to enable policy makers and public health advocates to better target policies and interventions to limit food and drink marketing and unhealthy food availability to children.

## Figures and Tables

**Figure 1 nutrients-12-02143-f001:**
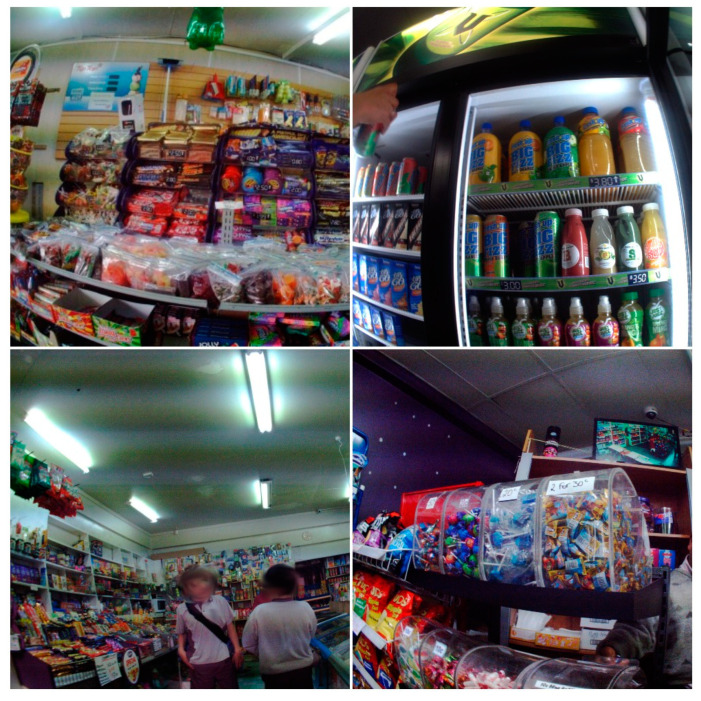
Noncore food and drink items available to children in convenience stores. Top left: single-serve confectionary including lolly-mix. Top Right: Sugary drinks in a branded refrigerator—note branding for “V” at the top and on shelves. Bottom right: single-serve confectionary note for price promotion of 2 for 30 c. Bottom left: image shows the “arms reach” of confectionary.

**Figure 2 nutrients-12-02143-f002:**
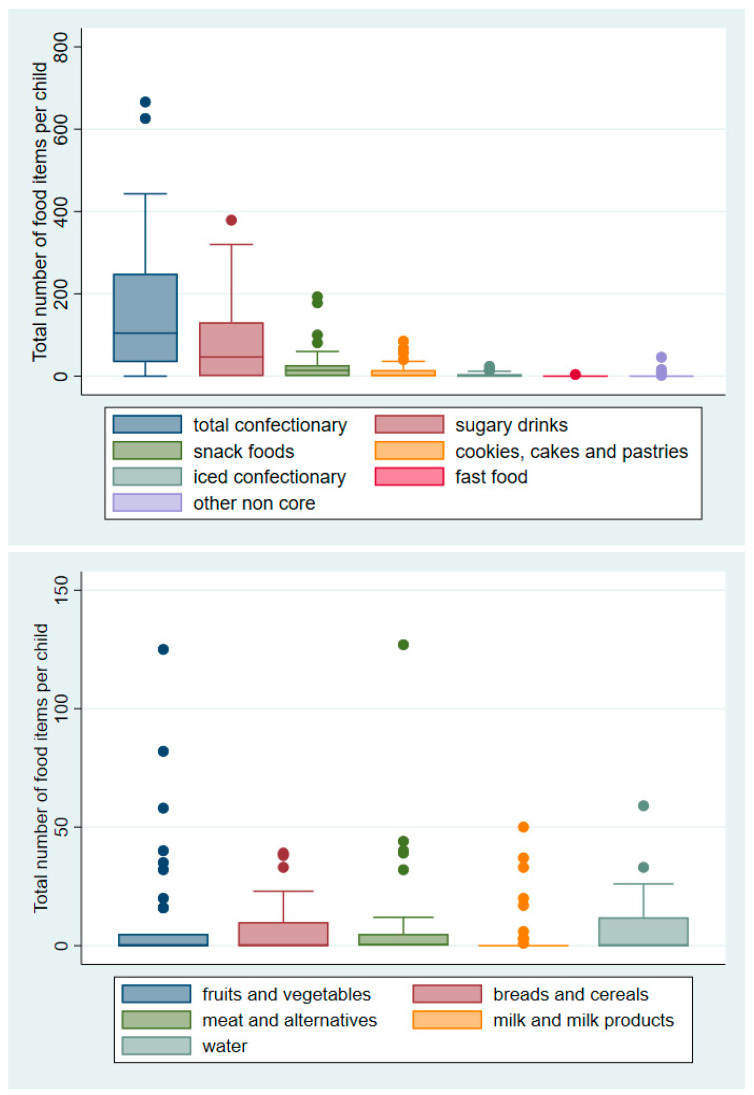
Median and interquartile range of noncore (top panel) and core (bottom panel) food and drink items available in convenience stores (Note the difference in axis between graphs for noncore and core food and drink. The dots represent outliers).

**Figure 3 nutrients-12-02143-f003:**
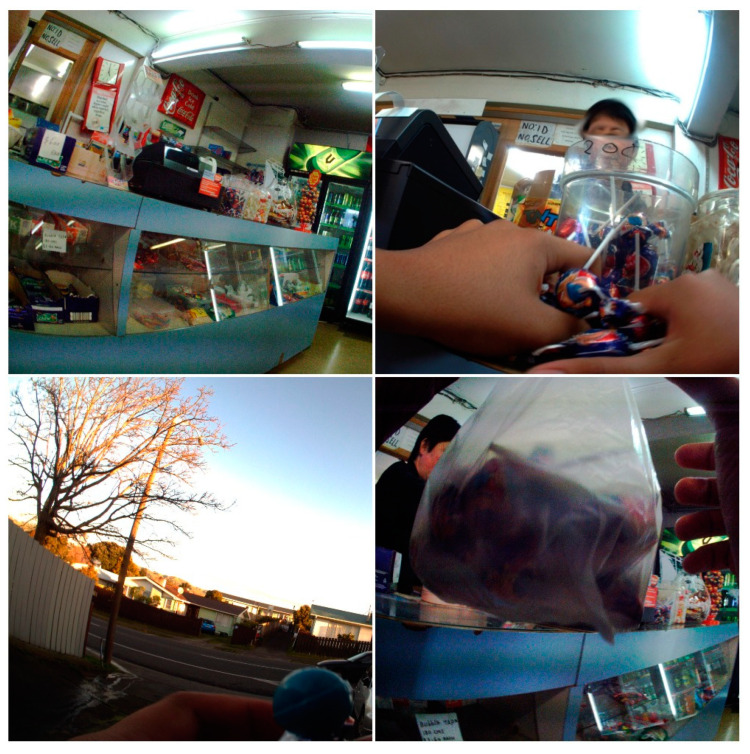
Food and drink purchase and consumption image sequence. Top left: countertop with confectionary displayed beside and beneath the counter. Top right: lollipops selected by participant for purchase. Bottom right: lollipops in a bag following purchase. Bottom left: consumption of lollipop.

**Table 1 nutrients-12-02143-t001:** Number of convenience store visits by a child.

Total number of visits	Frequency (*n*)	Percentage (%)
0	131	77.9
≥1	37	22
1	24	14.3
2	9	5.4
3	1	0.6
4	1	0.6
5	1	0.6
6	1	0.6
Total	168	100

**Table 2 nutrients-12-02143-t002:** Demographic characteristics of the Kids’Cam convenience store sample.

Demographic Variable		Conveniencestore Sample		Total Kids’Cam Sample	
		N	%	N	%
Total		37		168	
Gender	Male	18	48.6	80	47.3
	Female	19	51.4	88	52.7
Total		37		168	
Age *	11	4	10.8	13	8.0
	12	25	67.6	122	75.3
	13	8	21.8	26	16.1
	14	0	0	1	0.6
Mean	12.6				
Total		37		162	
Ethnicity	NZ European	15	40.5	66	39.3
	Māori	12	32.4	60	35.7
	Pacific	10	27.0	42	25.0
Total		37		168	
BMI category **	Underweight	2	5.4	9	5.4
	Healthy weight	21	56.7	87	52.1
	Overweight/ Obese	14	37.8	71	42.5
School Stratum ***	Low (Decile 1–3)	15	40.5	64	38.1
	Medium (Decile 4–7)	10	27.0	54	32.1
	High (Decile 8–10)	12	32.5	50	29.8

* Age missing for 6 participants (questionnaire not completed). ** BMI missing for 1 participant as child declined to be measured. *** Publicly funded schools in NZ are ranked by decile for funding purposes. Schools in decile 1 have the largest proportion of students from low socioeconomic backgrounds. Schools in decile 10 have the smallest proportion of these students [37].

**Table 3 nutrients-12-02143-t003:** Marketing exposure to food and drinks in convenience stores as means per child.

Marketing Medium	Core		Noncore		Difference		
	Mean	SD	Mean	SD	Mean	SD	95%CI
Packaging	1.7	1.7	7.9	4.3	6.1	3.9	4.9–7.4
Signs	1.4	1.5	6.8	4.9	5.4	4.2	4.0–6.8
Branded Display	0.6	0.8	4.2	3.0	3.6	2.7	2.8–4.5
Price Promotion	0.1	0.3	2.8	2.0	2.7	2.0	2.0–3.4

**Table 4 nutrients-12-02143-t004:** Purchases and consumption of food and drinks from convenience stores.

Items	Number purchased	%	Number Consumed	%
Total	74		33	
Noncore				
Confectionary (includes single serve)	22	29.7	7	21.2
Lolly-mix	8	10.8	6	18.2
Chocolate	3	4.0	1	3.0
Sugary drinks	20	27.0	9	27.3
Ice-cream	11	14.9	5	15.1
Pies	3	4.0	3	9.1
Cookies	1	1.4	1	3.0
Snack foods (potato crisps)	1	1.4	1	3.0
Other (noncore)	1	1.4	0	0
Total noncore	70	94.6	33	100

***Core***				
Milk	2	2.7	0	0
Bread	1	1.4	0	0
Water	1	1.4	0	0
Total core	4	5.4	0	0

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
