# Peer review of "Kids in a Candy Store: An Objective Analysis of Children’s Interactions with Food in Convenience Stores"

_nutrients, 2020, doi:10.3390/nu12072143_

Round 1
Reviewer 1 Report
This manuscript presents original, innovative and interesting data that could inform the public health debate around obesogenic environments. The study provides an interesting description of what convenience stores environments contain, from the children's point of view. Another dimension that is not considered here but could be discussed is affordability. It would be interesting as a future research perspective using that type of methodological approach.
I have some comments and suggestions for the authors.
I have two general (major) comments, and a number of more specific (minor) comments/questions/suggestions.
MAJOR COMMENTS
1/ My first main comment is that I am not certain that the methods and results match the objective announced in the introduction. The objective stated in the introduction was to examine "children's interactions with convenience stores" but at this point, it is not really clear what exactly was considered an interaction, and why interactions were an important thing to consider (see my more detailed comments below). The rest of line 60 indicates: "specifically: food and beverage availability, marketing, purchasing and
consumption" - I am guessing that these are what are considered interactions, but it should be clarified and justified (how are these variables considered as interactions?).
At the end of the methods section, it remains unclear what exactly was studied. It would be useful to define specifically, in the methods section, what was coded as an interaction and how. Was it the number of visits? The frequency? What was seen? what was bought? What was consumed? Or were all of these indicators used in order to describe the interactions? How was this analysed against the availability and access characteristics described at the beginning of the section? A more precise description of the analysis process would be helpful.
Again, the results section does not really revolve around interactions (in fact, the word "interaction" does not appear anywhere in this section), but rather around observation of availability and access, product placement etc. All those results are very interesting in themselves, I am just not certain that they allow to address the research objective as it is currently formulated.
2/ My second major concern is that this research is conducted on a small sample of children and while interesting, there are some issues regarding the generalization of the results. I think that this should be addressed more explicitly in the manuscript, to avoid over and mis-interpretations of the data presented.
MINOR SUGGESTIONS/COMMENTS/QUESTIONS
Lines 41-43: this statement needs some geographical contextualization. "Often located within children's neighborhoods" and "In many countries" are too vague. The studies cited are, respectively, in New Zealand, in London and in Ontario (Canada). Three English-speaking, developed countries with similar food cultures and also school meal systems. Is this also true of South American or EU countries? What about Asia?
I would therefore encourage the authors either to nuance their statement, or to provide proof that this statement about children's food environments holds true in the rest of the world.
Lines 47-48: Why is this topic important, beyond the fact that it has been identified in one article? I think the recent scholarly debate regarding the links between proximity of some food outlets and the eating habits/food purchases is worth mentioning - recent studies question this association (in adults), in particular because it does not take into account the fact that people may travel to get their food (see for instance Allcott, H., Diamond, R., Dubé, J. P., Handbury, J., Rahkovsky, I., & Schnell, M. (2019). Food deserts and the causes of nutritional inequality. The Quarterly Journal of Economics, 134(4), 1793-1844.) It may therefore be interesting to position the problem of children's food environments specifically within that debate.
Lines 47-52: please provide some justification for using this particular methodology to address this question. Why could it not be done with another more "classical" data collection method, what does this new approach allow to address that others do not? In particular, it would be useful to explain why an "objective analysis" is needed here.
lines 47-60: these two paragraphs could be rearranged to improve the flow - it is somehow confusing as the authors keep switching from research motivation to methodological considerations to context about NZ children and then back to more methodologies.
Line 56-57: As noted previously, there is no justification of why interactions between children and their food environment need to be studied - the fact that the FoodSee methodology allows it is not a justification.
Lines 62-70: though it may be published elsewhere, it would be useful in this paper to have the following information: 1) number of children in that area (to have an idea of what those 168 represent) 2) geographical distribution of these 168 children in the study area (since this is a study on environments)
Lines 71 -74: what was the definition of a convenience store to include the data? The information about how many children out of the 168 were included is only found on line 117 - it would be easier for the reader if it could be mentioned here. Likewise, the information of lines 142-143 regarding this subsample's characteristics could be presented here.
Line 78: the link to the protocol does not work, therefore I was unable to review it. Please provide at least a summary of main criteria. Is this a validated protocol? Was this published elsewhere?
Line 79-81: reading this, I am not sure whether you mean that each product type was counted (eg. if the image shows 2 chocolate bars of brand #1, and 5 chocolate bars of brand #2, you count "2") or that each product exemplary was counted (in the previous example, that would mean you count 7). Both are different measures of availability, and there are works in the field of choice architecture that show that the number of exemplaries has an impact on choice as well as variety does.
Line 81: for the sake of reproductibility, it would be interesting to know what it means that "care was taken". What measures were taken to ensure this, concretely?
Lines 96_98: to validate this definition of "accessible", it would be useful to know the variance of height between the children included, and to see whether they are representative of the heights of NZ children this age (I'm guessing that there are some rather large variations between 11 and 14 years old, and between boys and girls). My concern here,if the sample is too small and not representative in terms of height, is that it is possible that some items were marked as accessible to the recording children, but that these children are especially tall (or small) which means that the accessibility is overrepresented in the sample (or vice versa). Maybe the authors could provide some details as to how this was addressed/considered in the study?
Lines 118-153 seem to me to belong to the results section rather than the methods
Lines 174-176: I think this creates ambiguity and could potentially lead to overinterpretation. Please explain what it means, precisely, and concretely, that "caution should be taken in interpretation". Does this mean that this result is not statistically robust (and therefore, why formulate it as a significant difference?) or that it can be interpreted (and in that case, please indicate what can be said about it). The same goes by line 172: if we have a problem of sample representativeness, then how can we say that there are similarities across ethnicities (or what can we really say about that)
Line 179: shouldn't it say "non-core" instead of "non-care"?
Figure 2 : are packages of several units (eg. packs containing two or three chocolate bars) counted as one unit in this graph? I find it hard to compare the amount of products across categories without knowing the units/portions within those categories. Especially regarding fruits and vegetables in the second graph: what is the unit representing "total food"?
The results of the study are interesting in themselves, and the discussion does a rather good job of highlighting the important findings. However, the discussion should insist on the domain of validity of the results. The subsample was small and lots of representativeness biases (some mentioned in the results sections, some not) need to be listed. In the same line, I believe that the conclusions should be nuanced by giving some perspective on the extent to which the findings are transferrable to other geographical areas (cities, countries) and populations and cultures; and on the importance of convenience stores as sources of dietary intake in this population and others.
Reviewer 2 Report
This is an important study that adds to the literature describing the current retail food environment and consumer behavior. It adds to the growing evidence-base that in-store marketing, including product, price, placement, and promotion strategies, influences the food and beverages shoppers purchase and eat. The ubiquity of unhealthy food and beverages in convenience stores and the ways they are aggressively marketed in this setting contribute to the purchase and consumption of unhealthy options. These and similar findings necessitate a response from policymakers and public health advocates about changes that should be required to improve the healthfulness of the retail food environment, including increasing marketing of healthy food and beverages and decreasing marketing of unhealthy options.
In addition to describing the current retail food environment, this study examines consumer behavior. It is unique in that it not only studies consumer purchasing behavior, but also marketing exposure and consumption. Studying consumption was possible because the study utilized data collected using wearable cameras, an innovative research tool that allowed the participants to be studied as they went about their day-to-day lives, which is likely more accurate than food diaries or recall. It is possible, however, that children and their families behaved differently while wearing these cameras than at other times.
It would be helpful to understand the nutrition criteria used for core versus non-core food and beverages, and to better understand the classification of beverages in the study (the link provided in the paper did not function for us). For example, were 100 percent fruit juices categorized as core or non-core? Did they discern between flavored waters with and without added caloric sweeteners?
On page five, lines 169-171, please provide more clarity on the types of core foods available. Was the meat, jerky or meat sticks? Was the fruit dried with added sugars or a banana?
In section 3.2 Placement of food and drink items by category, it would be interesting to contrast the results with results for core food and beverages.
In addition to the approaches described in the paper on page 11, other policy approaches to address the current retail food environment could be added at lines 313-314, such as “removing them from countertops and areas above and below the cash registers (at checkout) and end-of-aisle shelves.” Also note that sugar-sweetened beverages could be grouped together in a common aisle and section of the coolers, to allow people to more easily identify them and avoid them.
I agree with the authors’ recommendation that a similar study should be conducted about the current retail food environment and consumer behavior in supermarkets. Only 22 percent of the children in the study visited a convenience store during the study period. While improving the retail food environment in convenience stores is important, a public health approach to reducing childhood obesity calls for the improvement of all retail food environments.
Round 2
Reviewer 1 Report
Thank you for your thorough response. This is a very good paper!